# Biobanking Organoids or Ground-State Stem Cells?

**DOI:** 10.3390/jcm7120555

**Published:** 2018-12-16

**Authors:** Wa Xian, Marcin Duleba, Yusuke Yamamoto, Matthew Vincent, Frank McKeon

**Affiliations:** 1Institute of Molecular Medicine, McGovern Medical School of University of Texas Health Science Center, Houston, TX 77030, USA; 2Department of Biochemistry and Molecular Biology, University of Texas McGovern Medical School, Houston, TX 77030, USA; 3Department of Biology and Biochemistry, University of Houston, Houston, TX 77204, USA; marcin.duleba@gmail.com (M.D.); fdmckeon@uh.edu (F.M.); 4Division of Molecular and Cellular Medicine, National Cancer Center Research Institute, Tokyo 1040045, Japan; yusukeyamamoto198176@gmail.com; 5Tract Pharmaceuticals, Inc., Marlborough, MA 01752, USA; mvincent@tractpharma.com

**Keywords:** adult stem cells, ground-state, autologous transplantation, regenerative medicine, biobanking

## Abstract

Autologous transplantation of human epidermal stem cells cultured in Green’s method is one of the first examples of utilizing adult stem cells in regenerative medicine. Using the same method, we cloned p63-expressing distal airway stem cells and showed their essential role in lung regeneration in a mouse model of acute respiratory distress syndrome. However, adult stem cells of columnar epithelial tissues had until recently evaded all attempts at cloning. To address this issue, we developed a novel technology that enabled cloning ground-state stem cells of the columnar epithelium. The adaption of this technology to clone stem cells of cancer precursors furthered our understanding of the dynamics of processes such as clonal evolution and dominance in Barrett’s esophagus, as well as for testing platforms for chemical screening. Taken together, the properties of these ground-state stem cells, including unlimited propagation, genomic stability, and regio-specificity, make them ideal for regenerative medicine, disease modeling and drug discovery.

## 1. Introduction

One of the most exciting directions in medicine is based on stem cell research, which holds the promise of treatments and cures for various diseases and conditions that have so far frustrated traditional pharmaceutical approaches. Both stem cell research and clinical trials in regenerative medicine are presently dominated by pluripotent stem cells: embryonic stem cells (ESC) and induced pluripotent stem cells (iPSCs). However, these remarkable cells face important and unresolved hurdles, including the risk of generating teratomas, the truly arduous and inefficient processes of directed commitment to desired lineages and the limited regenerative capacity of derived lineages [1,2,3,4,5,6]. The promise of pluripotent stem cells has largely overshadowed efforts to harness so-called “adult” or “somatic” stem cells intrinsic to regenerative tissues. Green and colleagues developed methods for cloning epidermal stem cells [7] in their most immature and clonogenic form, which could be differentiated at will to yield a stratified squamous epithelium. The Green strategy has also been applied to capture stem cells from other stratified epithelia, including those of the corneal, thymic and airway epithelia [8,9,10]. However, stem cells of columnar epithelial tissues, such as those of the gastrointestinal tract, liver and kidney, have resisted cloning in a manner that maintains their immaturity during proliferative expansion. Instead, many researchers maintain these columnar epithelia as regenerative “organoids” that contain a minor fraction of stem cells that drive their growth [11,12,13]. While their potential for regenerative medicine is obvious [14], organoids are labor-intensive, very slow to expand and mostly comprised of differentiated cells rather than regenerative stem cells. To overcome this significant barrier in adult stem cell biology, highly robust technologies have been developed to clone and propagate clonogenic, “ground-state” stem cells of the human intestine and colon [15]. These cultured stem cells show remarkable stability in their genomic integrity and epigenetic commitment programs, show unlimited replicative expansion and maintain high clonogenicity, suggesting tremendous potential in disease modeling and regenerative medicine.

## 2. Technologies for Adult Epithelial Stem Cell Culturing

Since 1975, the 3T3J2 feeder system generated in Green’s lab has been used widely to culture adult stem cells derived from p63-expressing stratified epithelium, including skin, thymus and lung [7,9,10]. Recently, a new technology developed in our lab, using a combination of 3T3J2 feeder cells and a specialized medium, was employed to clone ground-state stem cells from the columnar epithelium [15,16,17]. The common feature of these two feeder-dependent systems is their ability to keep highly clonogenic adult stem cells in culture so that stem cells can rapidly proliferate and reach sufficient numbers required for downstream applications, such as autologous transplantation, in a short time. In addition, the adult stem cells cultured in these feeder systems maintain the unipotent (e.g., skin) or multipotent differentiation ability (e.g., lung and intestine) and region-specificity despite long-term culturing [7,10,15,18]. The J2 strain of 3T3 cells, originally developed by Green, was used in both of these systems and cleared regulatory hurdles for clinical use thirty years ago, without reports of adverse effects [19,20,21,22,23,24]. Several clinical-grade cell culture methods have been developed under feeder-free conditions [20,21,22]. However, methods based on the Rheinwald and Green culture protocol remain the gold standard for clinical applications [23] due to the high clonogenicity and regenerative capacity.

Due to the different signaling requirements of the maintenance of stemness in the stratified epithelium and columnar epithelium, the growth medium of Green’s method and the method that we developed (herein Xian–McKeon method) are significantly distinct. In addition to the growth factors such as IGF and EGF that were included in the medium in Green’s method, the Xian–McKeon lab developed a media containing novel combinations of growth factors and regulators of TGF-β, Wnt/β-catenin, EGF, IGF, and Notch pathways [25,26] that supports the maintenance of human columnar epithelial stem cells, including intestinal stem cells, in a highly clonogenic, ground-state form. Importantly, single-cell derived, pedigree lines of human intestinal stem cells can be induced to differentiate into all cell types of the intestinal epithelium in air–liquid interface culture systems (Figure 1).

Regio-specificity of adult stem cells has been demonstrated in both of these systems [10,15]. In the Xian–McKeon method, the ground-state stem cells possessed robust epigenetic programs of commitment to regio-specific intestinal differentiation that are stable, despite more than a year of continuous propagation. This cell-autonomous regio-specificity of stem cells along the intestinal tract argues against a unitary “intestinal stem cell” or even one for each of the histologically recognized segments, and rather demonstrates a developmentally established spectrum of stem cells that ultimately maintain the histological and functional properties that define these segments. A heuristic deciphering of the commitment code from the regio-specific expression patterns described in Wang et al. [15] will guide parallel efforts with iPSCs to achieve appropriate lineage fates.

Despite the extensive studies on p63-expressing stem cells in the upper airways [27] and our own findings of p63-expressing cells contributing to lung regeneration following acute respiratory distress syndrome in mice [10,18], it was completely unclear whether we were dealing with one p63+ airway stem cell occupying different niches or many that were committed to regio-specific differentiation. Green’s method was used to clone tracheal airway stem cells (TASCs) and distal airway stem cells (DASCs). These regio-specific airway stem cells demonstrated a clear distinction in their differentiation ability in various in vitro assays and in vivo transplantation experiments. Thus, despite the morphological and gene expression similarities between the upper airway stem cells and the distal airway stem cells, they clearly possess distinct gene expression profiles and cell fate programs. However, in the final analysis, the identification of distinct stem cells in the airways is of critical importance for any future scheme in regenerative medicine or the testing of drugs or biologics for their ability to rally such cells for regenerative endpoints.

Human ESC and iPSC lines acquire, with successive passages, genomic structural variations, including some that confer a selective advantage [28,29]. To assess the genomic stability of ground-state stem cells, we examined the copy number (CNV) and single nucleotide variation (SNV) in our cloned ground-state intestinal stem cell pedigrees and showed that ground-state stem cells sustain few genomic changes within the first 100 days of proliferative expansion [15]. Due to their high clonogenicity, we estimated that more than one billion stem cells would be generated from a single stem cell following 60 days culturing in this system [17]. Previous research suggested that epithelial cells require the loss of the pRB/p16^INK4a^ cell cycle control mechanisms, in addition to hTERT overexpression, to achieve immortality [30]. However, the remarkable proliferative potential of these ground-state stem cells, despite the absence of RB or p16 mutations suggests that their apparent immortality is a normal and intrinsic property of adult stem cells, rather than an indication of pathology.

## 3. Ground-State Stem Cells versus Organoids

Mammalian cells have been cultured in collagen and laminin-rich matrices as organoids in 3D culture since at least 1980 [12,31,32,33,34,35]. An organoid is defined as a unit of function of a given organ that is able to reproduce, in vitro, a biological structure similar in architecture and function to its counterpart in vivo. There are multiple origins of organoids that include a fragment of tissue, a stem cell isolated in an adult organ, an embryonic stem cell, or an induced pluripotent stem cell. Importantly, organoids can also be generated from cells that have lost stem cell capacity—so-called “transit amplifying cells” with limited proliferative potential—indicating that organoids per se are not strictly assays for stem cells [36].

Despite fundamental advances in stem cells afforded by organoid technology [37], the organoid method does not support cloning the intestinal stem cells or any other adult stem cells in their highly immature, clonogenic state. We solved the problem of culturing populations of pure stem cells of the human gastrointestinal tract using a cocktail of factors impacting Wnt, Notch, and TGF-β signaling pathways among others and were able to clone ground-state stem cells from all regions of the intestine and colon from endoscopic biopsies [15,17]. In other words, the stem cells of columnar epithelia can now be captured in the same ground-state that Green showed for the stem cells of stratified epithelia such as the epidermis. We found the following properties of these ground-state stem cells that make this technology particularly advantageous compared with the organoid method: (1) a typical 1 mm endoscopic biopsy yields 100–300 independent stem cell clones; (2) each of these clones can be independently propagated while maintaining genomic integrity; (3) these clones uniformly express stem cell markers and have a clonogenicity rate of 60–90%; (4) each of these clones retain epigenetic memory of the region from which they were derived; and (5) single clones can be differentiated to yield *3-D* intestinal epithelia with regionally-appropriate cell types.

It should be acknowledged that many of the features of these gastrointestinal stem cells were unknown prior to Wang et al. [15], Yamamoto et al. [16] and Duleba et al. [17], including the stability of the epigenetic programs underlying commitment, despite months of continuous in vitro propagation, the fact that these stem cells possess all of the information required to assemble into a higher order, and that regio-specific *3-D* epithelia are nearly identical to those found in the mature intestine in vivo. As stem cells comprise only a minor component of organoids, perhaps less than 1% [38], the molecular features of the stem cells of columnar epithelia, such as the intestinal tract, have remained unclear (Figure 2). Therefore, the selective cloning and proliferative expansion of highly clonogenic, ground-state intestinal stem cells through the Xian–McKeon method offer a first glimpse into the molecular properties of these cells. Our recent success in adapting this technology to clone stem cells in precancerous lesions, such as Barrett’s esophagus, addressed the long-standing confusion in the cellular origin of Barrett’s esophagus. Furthermore, using this technology, we generated broad sets of patient-matched stem cells corresponding to all stages of precursor lesions both in-line and out-line with the progression to adenocarcinoma. We can anticipate that each of these clones can be highly annotated with genomics, gene expression and differentiation fate information, accessible through a dynamic biorepository and thus be valuable substrates for new investigations across the cancer biology community.

## 4. Biobanking of Ground-State Stem Cells and Personalized Regenerative Medicine

The onset of adult-stem-cell-based regenerative medicine started in the 1980s. Green and colleagues demonstrated the first example of cell therapy using cultured stem cells. They showed that human epidermis could be grown in the laboratory and transplanted to patients to reconstitute a functional epidermis [39,40]. Since then, transplantation of cultured epidermal stem cells has long been used to treat patients with burns, chronic wounds and stable vitiligo [41]. This is a life-saving procedure for patients with large area of burns. Moreover, the long-term effectiveness and safety of genetically-modified epidermal stem cells in correcting the severe skin blistering disease, epidermolysis bullosa, has been shown clinically [42].

In addition to skin, cultured stem cells from other epithelial tissues can be the source of stem-cell-based regenerative medicine. For example, a feature of lung regeneration that bodes well for regenerative medicine is that the underlying stem cell is highly clonogenic, which shows unlimited expansion capacity in vitro, and readily transplants to form functional alveoli in acutely damaged lungs [18]. We showed that a single p63+/Krt5+ DASC can be cloned, expanded and transplanted via intratracheal delivery to acutely damaged lungs, where they selectively inhabit damaged regions and differentiate to form Clara cells and alveoli composed of type I and type II pneumocytes. Importantly, these same p63+/Krt5+ DASCs showed no incorporation in mice without prior acute lung injury, suggesting that the efficient regenerative properties of these cells are not marred by “off-target” incorporation. Lastly, DASCs are readily cloneable from simple bronchoscopic biopsies, from bronchopulmonary lavage, or from transmural biopsies, providing good sources of autonomous stem cells that can be expanded to hundreds of billions of cells in weeks. Taken together, the established properties of DASCs, including clonogenicity, expandability, and facility for accurate transplantation obviate many theoretical objections that could have limited their use in regenerative medicine for either acute or chronic lung diseases.

It is also conceivable that the newfound ability to derive ground-state stem cells from columnar epithelium provides excellent sources for autologous transplantation to treat a wide range of disorders that current treatments are not able to help. For example, cultured ground-state intestinal stem cells may be of use to restore the intestinal epithelial functions following autologous transplantation in patients with severe forms of short bowel syndrome (SBS) [43], congenital disorders [44], or inflammatory bowel disease (IBD) [45,46].

In conclusion, we now have the technologies for cloning and culturing adult stem cells from nearly all types of epithelial tissues. Given the tremendous success of the pioneering work by Green and colleagues on the use of cultured adult stem cells in regenerative medicine, we should anticipate that more scientists and clinicians will recognize the potential of adult stem cells, appreciate the potential of biobanking various types of adult stem cells from individuals of diverse HLA haplotypes, and make attempts to use them for stem-cell-based personalized regenerative medicine.

## Figures and Tables

**Figure 1 jcm-07-00555-f001:**
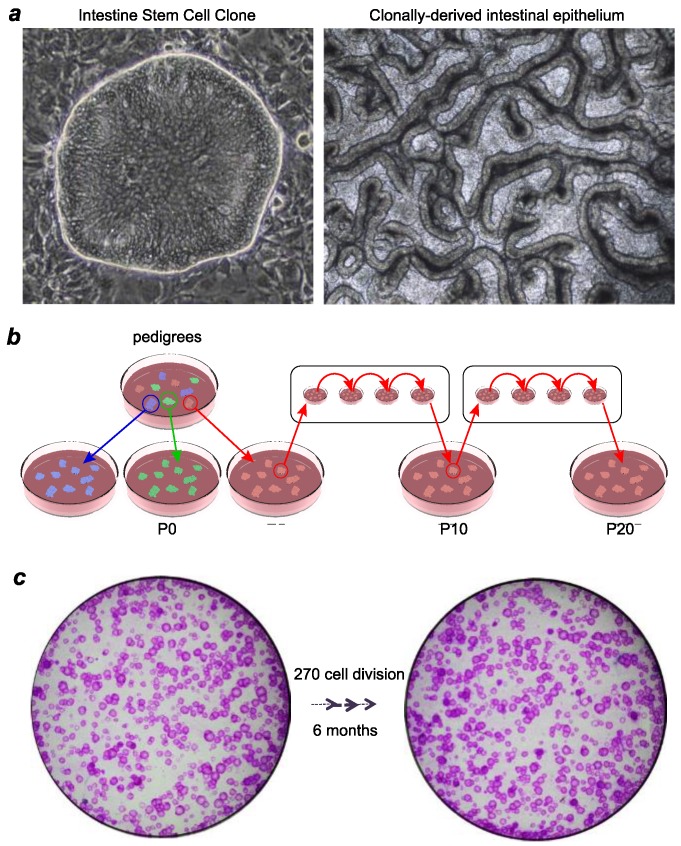
Xian–McKeon method of cloning ground-state intestinal stem cells. (**a**) Right, Representative image of a single-cell derived ISC^GS^ colony. Left, Top view of in vitro intestinal epithelium generated from stem cells of a pedigree of ISC^GS^ differentiated in an air–liquid interface setting. (**b**) Schematic diagram of establishing pedigrees from ISC^GS^. Long-term culturing does not affect the capacity of ISC^GS^ to form single-cell derived pedigrees. (**c**) Clonogenicity assay revealing nearly unchanged number of Rhodamine red-stained colonies despite long-term culturing.

**Figure 2 jcm-07-00555-f002:**
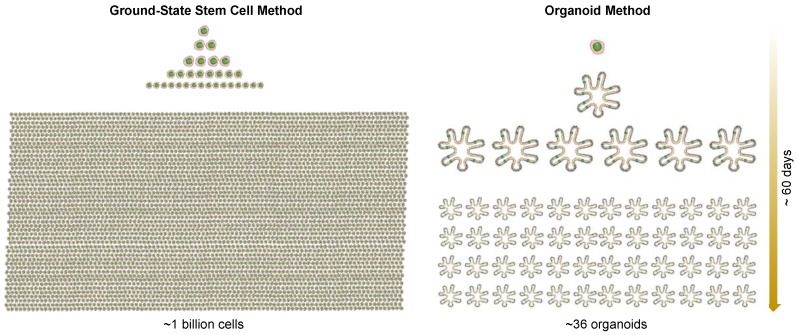
Rapid expansion of a single cell to one billion cells in approximately 60 days using the Xian–McKeon method. In comparison, one cell can become 36 organoids in the organoid method.

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
