# Peer review of "Biobanking Organoids or Ground-State Stem Cells?"

_jcm, 2018, doi:10.3390/jcm7120555_

Round 1
Reviewer 1 Report
In this manuscript the Authors dealt with an important issue in regenerative medicine and tissue engineering, i.e. the development of methodologies aimed at obtaining and maintaining an epidermal stem cell population in vitro and its involvement in further studies like regenerative medicine, but also disease modeling and drug discovery purposes.
The review abstract and the article are mainly related to the research work of the Authors, though. In fact, the reader would expect a true research original paper after the abstract. A lot of interesting scientific details are reported from papers of the Authors but there fewer reference about this argument from other research groups. The review must have a wider field of interest as the reader would expect.
Finally, I suggest to change the title of “2. Technologies for Adult Stem Cell Culturing” because in the text the Authors talk about only epidermal stem cells technologies and do not cover technologies for others adult stem cells (e.g. mesenchymal stem cells). Moreover, there are few typos in the text (for example page 3 line 113).
Please add a reference related to the veterinary world:
Equine epidermis: a source of epithelial-like stem/progenitor cells with in vitro and in vivo regenerative capacities.
Broeckx SY, Maes S, Martinello T, Aerts D, Chiers K, Mariën T, Patruno M, Franco-Obregón A, Spaas JH.
Stem Cells Dev. 2014 May 15;23(10):1134-48. doi: 10.1089/scd.2013.0203.
Author Response
In this manuscript the Authors dealt with an important issue in regenerative medicine and tissue engineering, i.e. the development of methodologies aimed at obtaining and maintaining an epidermal stem cell population in vitro and its involvement in further studies like regenerative medicine, but also disease modeling and drug discovery purposes.
The review abstract and the article are mainly related to the research work of the Authors, though. In fact, the reader would expect a true research original paper after the abstract. A lot of interesting scientific details are reported from papers of the Authors but there fewer reference about this argument from other research groups. The review must have a wider field of interest as the reader would expect.
This review paper by Xian and colleagues focuses on the difference between widely-used organoid culture systems and an emerging technology of cloning and expanding ground-state epithelial stem cells. The manuscript thoroughly describes the successful use of epidermal stem cells culturing on 3T3 feeder cells in clinical settings for regenerative medicine and the history of Green method. In addition, the review describes in depth the problems that organoid culture systems are facing and explains the advantages and limitations of ground-state stem cell cloning technology. One of the goals of this review paper is to bring readers' attention to the existence of this revolutionary technology generated in the authors’ laboratories. Since the technology is still new and the authors need to use their own work as examples in this review article.
Finally, I suggest to change the title of “2. Technologies for Adult Stem Cell Culturing” because in the text the Authors talk about only epidermal stem cells technologies and do not cover technologies for others adult stem cells (e.g. mesenchymal stem cells). Moreover, there are few typos in the text (for example page 3 line 113).
We changed the title to “Adult Epithelial Stem Cell Culturing”.
Please add a reference related to the veterinary world:
Equine epidermis: a source of epithelial-like stem/progenitor cells with in vitro and in vivo regenerative capacities.
Broeckx SY, Maes S, Martinello T, Aerts D, Chiers K, Mariën T, Patruno M, Franco-Obregón A, Spaas JH.
Stem Cells Dev. 2014 May 15;23(10):1134-48. doi: 10.1089/scd.2013.0203.
We have added this reference in the manuscript.
Reviewer 2 Report
In this manuscript, Xian et al. overviewed a novel culture techonology for epithelial stem cells and the potential of these stem cells to regenerative medicine. This manuscript is very interesting and well-writtened. I request some minor revision, as follows.
It is better to discriminate between author's and other's studies. For example, Xian and Mckeon laboratories (P2-L53, L68) are author's group.
aP2-L69, there are some extra spaces.
There are various abbrevations that never appear at later parts.
P3-L124, TGF-beta is inconsistent with previous description.
Author Response
In this manuscript, Xian et al. overviewed a novel culture techonology for epithelial stem cells and the potential of these stem cells to regenerative medicine. This manuscript is very interesting and well-writtened. I request some minor revision, as follows.
1. It is better to discriminate between author's and other's studies. For example, Xian and Mckeon laboratories (P2-L53, L68) are author's group.
We made clarifications at line 101.
2.aP2-L69, there are some extra spaces.
We modified it.
3.There are various abbrevations that never appear at later parts.
We corrected it.
4.P3-L124, TGF-beta is inconsistent with previous description.
We corrected it.